# Cover-separable Fixed Neural Network Steganography via Deep Generative Models

## ABSTRACT

Image steganography is the process of hiding secret data in a cover image by subtle perturbation. Recent studies show that it is feasible to use a fixed neural network for data embedding and extraction. Such Fixed Neural Network Steganography (FNNS) demonstrates favorable performance without the need for training networks, making it more practical for real-world applications. However, the stego-images generated by the existing FNNS methods exhibit high distortion, which is prone to be detected by steganalysis tools. To deal with this issue, we propose a Cover-separable Fixed Neural Network Steganography, namely Cs-FNNS. In Cs-FNNS, we propose a Steganographic Perturbation Search (SPS) algorithm to directly encode the secret data into an imperceptible perturbation, which is combined with an AI-generated cover image for transmission. Through accessing the same deep generative models, the receiver could reproduce the cover image using a pre-agreed key, to separate the perturbation in the stego-image for data decoding. such an encoding/decoding strategy focuses on the secret data and eliminates the disturbance of the cover images, hence achieving a better performance. We apply our Cs-FNNS to the steganographic field that hiding secret images within cover images. Through comprehensive experiments, we demonstrate the superior performance of the proposed method in terms of visual quality and undetectability. Moreover, we show the flexibility of our Cs-FNNS in terms of hiding multiple secret images for different receivers.

## CCS CONCEPTS

• **Information systems** → *Multimedia information systems*; • **Security and privacy** → *Security services*.

## KEYWORDS

Image steganography, Fixed neural network, Separable perturbation

## 1 INTRODUCTION

Image steganography aims to hide a form of secret data within a cover image for covert communication, where only the informed receiver with a shared key is able to extract the secret data. To avoid being detected by human or machine eavesdroppers, the stego-image (i.e., the image with hidden data) should be visually and statistically indistinguishable from the cover image. Earlier image steganographic methods hide the secret data by altering

*ACM MM, 2024, Melbourne, Australia*

the least significant bits of the pixels in the cover image. Later, studies follow the Syndrome-Trellis Codes (STCs) framework [8] to minimize the distortion caused by data embedding.

Like many fields in signal and image processing, steganography is revolutionized by the remarkable development of deep neural networks (DNNs). A typical steganographic technique with DNNs is learned neural network steganography (LNNS), which transforms the handcrafted conventional steganography into a data-driven manner [1, 20, 29, 32, 41, 45, 49]. LNNS follows an autoencoder approach with two key components: a secret encoding network to hide the secret data into a cover image, and a secret decoding network to recover the secret data from a stego-image. These two networks need to be jointly learned to minimize hiding distortion and recovery error. Despite simultaneously achieving good visual quality with high payload capacity, the LNNS schemes require a large amount of data and computational resources to train well-performing steganographic networks (i.e., the secret encoding or decoding networks). On the other hand, the well-trained steganographic networks, whose size are relatively large, must be covertly transmitted to the sender or receiver who does not possess any steganographic tools [7, 25, 26].

To avoid training and transmitting the steganographic networks, researchers propose Fixed Neural Network Steganography (FNNS) [10, 23, 30], which abandons the learned encoding network and conducts data hiding and recovery using a fixed decoding network. This technique keeps the parameters of the decoding network fixed and alters the cover image with adversarial perturbations [12, 36] so that the modified cover image (i.e., the stego-image) could trigger the decoding network to output the secret data. At the receiving end, the secrets can be extracted by inputting the stego image into the same decoding network. Compared to LNNS, FNNS does not require network training. Moreover, it could use random decoding networks with initialized weights to hide and extract secret data, allowing the sender and receiver to synchronize steganographic tools by sharing the architecture of the fixed decoding network and the random seed used to initialize its weights. However, the stego-images generated by the existing FNNS methods exhibit high distortion, making them easily detected by steganalysis tools. In high-capacity scenarios, such as 4 bits per pixel (bpp), the distortion can even be perceptible to the human eye.

To address the issue of the FNNS, we propose in this paper a Cover-separable Fixed Neural Network Steganography (Cs-FNNS) by exploiting the advantage of the deep generative models [16, 17] for content generation, as shown in Fig. 1. Our Cs-FNNS is motivated by the fact that AI-generated content is widespread over the internet, transmitting and sharing such data would be considered to be normal without causing suspicion. Cs-FNNS searches an invisible perturbation that could trigger the fixed random decoding network to output the secret data and combines it with a cover image to produce the stego-image. The cover image is AI-generated and always remains consistent when the inputs (i.e., noise maps

generated by keys) of the deep generative model are the same. In data extraction, the receiver first uses the shared key to reproduce the cover image to separate the perturbation in the stego-image, which is sent to the same decoding network to recover the hidden data.

In contrast to the previous FNNS methods [10, 23, 30], which directly extract secret data from the stego-image, we separate the decoding of the secrets from the cover image. Such a strategy eliminates the disturbance of the cover image on the decoding network, encouraging Cs-FNNS to search for smaller perturbations, which introduce lower distortion to the stego-images. To adapt to this novel decoding way, we propose a Steganographic Perturbation Search (SPS) algorithm to directly encode the secret data into imperceptible perturbations. By using the SPS, we could not only find minor perturbations that make a single decoding network produce the intended output, but also special ones that could trigger different decoding networks to output different secret data. Following [2, 20, 29], we instantiate our Cs-FNNS for hiding secret images into cover images. Through comprehensive experiments, we indicate the superior performance of the proposed method compared with the state-of-the-art (SOTA) FNNS schemes in terms of visual quality and undetectability.

Our main contributions are summarized below:

- We explore the possibility of incorporating the AI-generated content (AIGC) to boost the performance of the FNNS. By leveraging the capability of the AIGC for content generation, we propose Cs-FNNS to separate the decoding of the secrets from the cover image, hence achieving a better performance.
- We propose a Steganographic Perturbation Search (SPS) algorithm to directly encode the secret images into imperceptible perturbations, which brings negligible distortion into the stego-images.
- Experiments demonstrate that the proposed method is 1) convenient, without training or transmitting the secret steganographic encoding and decoding networks, 2) secure, generating high-quality stego-images that are able to fool the SOTA steganalysis tools, and 3) flexible, hiding multiple secret images for different receivers in a single cover image.

## 2 RELATED WORK

### 2.1 Traditional Image Steganography.

Traditional image steganography designs hand-craft algorithms to modify the cover image for data hiding, which could be briefly categorized into spatial domain-based steganography [6, 31] and transformation domain-based steganography [38]. The former directly changes the pixel values in the spatial domain, while the latter alters the coefficients in the transform domain to accommodate the secret data. To enhance the undetectability of the stego-images, researchers propose adaptive image steganography, which can be adopted to perform data hiding in the spatial or transformed domain [18]. The adaptive methods are executed under a distortion-coding framework, aiming to minimize a particular distortion function caused by data hiding. The most famous framework for adaptive steganography is proposed in [8], where the Syndrome-Trellis Codes (STC) is utilized to encode the secrets. Similarly, some other adaptive methods are designed with different distortion functions [18, 19, 24]. In general, the capacity of these schemes has to be limited for high undetectability ($\leq$0.5bpp).

### 2.2 Learned Neural Network Steganography.

Encouraged by data-driven deep learning techniques, recent works propose learned neural network steganography (LNNS) to train deep encoding and decoding networks for data hiding and recovery. Hayes et al. [14] pioneer the research of such a technique, where the secret data are concealed into a cover image or recovered from a stego-image using an end-to-end learnable DNN. Zhu et al. [49] insert adaptive distortion layers between the encoding and decoding networks to improve the robustness. Baluja [1, 2] firstly proposes to conceal a whole color image into another one for large capacity data hiding, where a preparation network is designed to transform the secret image into feature maps before data hiding. Based on the work in [1], Rahim et al. [32] add a penalty loss on the weight of the steganographic networks to stabilize training. Zhang et al. [45] propose a universal framework to transform the secret image into invisible high-frequency components for data hiding. The latest studies explore invertible networks to hide images within images [13, 20, 29, 39], where the forward and backward computation processes of the invertible network serve as the encoding and decoding networks, respectively. LNNS achieves high capacity with considerable visual quality. However, it requires training and covertly transmitting the steganographic networks.

### 2.3 Fixed Neural Network Steganography.

Recent advances [10, 23, 30] propose Fixed Neural Network Steganography (FNNS), which leverages the neural network's sensitivity to minor perturbations to hide secret data. They modify the cover image with adversarial perturbation [12, 36] to obtain the stego-images that could trigger the fixed decoding network to produce some specific outputs corresponding to the secret. Ghamizi et al. [10] generate the stego-images by encoding the secret data into image labels, the capacity of which is rather limited. Kishore et al. [23] increases the capacity by augmenting the dimensions of the output of the decoding network, and use a message loss to produce the stego-images. Luo et al. [30] propose a key-based FNNS scheme to prevent unauthorized data recovery, where a key-controlled perturbation is added on the cover images before data embedding. Unfortunately, the stego-images generated by the existing FNNS schemes show high distortion and are prone to be detected by steganalysis tools, In high-capacity scenarios (i.e., >4bpp), the stego-images by them exhibit noticeable noise, which can be easily perceived by the human eye.

### 2.4 Deep Generative Models.

In recent years, deep generative models, such as generative adversarial networks (GANs), variational autoencoders (VAEs), flow models [22], and diffusion models [16], have emerged and flourished. They are neural networks trained on massive datasets and can approximate the intricate, high-dimensional probability distribution of the training samples. Deep generative models have been widely used for creating the AIGC and successfully applied in numerous domains, including computer vision [17], natural language processing [5], privacy preserving [43], and even biosciences

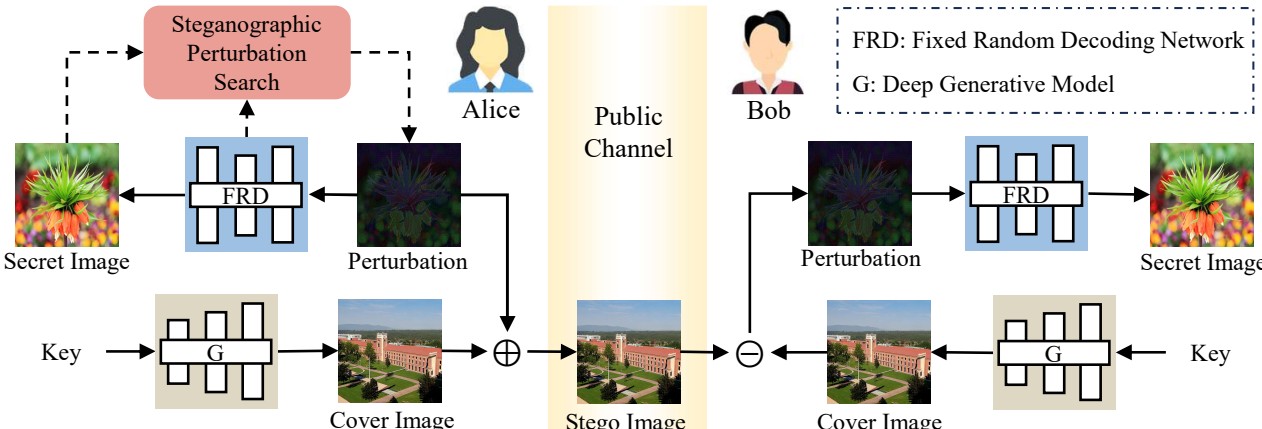

**Figure 1: Cs-FNNS workflow: Alice (sender) uses the proposed Steganographic Perturbation Search Algorithm to find a perturbation that makes the fixed random decoding network output the secret image, then adds the perturbation on an AI-generated cover image (controlled by a key) to produce a stego-image; Bob (receiver) first uses the shared key to reproduce the cover image to separate the perturbation in the stego-image, then decodes the secret image with the same decoding network.**

[44]. Rombach *et al.* [33] make use of diffusion models to synthesize high-resolution images. Brown *et al.* [5] train an immensely large language generation model, GPT-3, which is capable of text summarizing, editing and composing, or providing assistance for programming. Yuan *et al.* [43] take advantage of the GAN to generate identifiable virtual faces for privacy preserving. Zeng *et al.* [44] utilize deep generative models to explore and navigate the vast molecular space of drugs, so as to expedite the drug design process.

In this paper, we explore the possibility of using the existing AIGC techniques to facilitate steganography. With the help of AIGC techniques, we transform the existing Fixed Neural Network Steganography (FNNS) into a cover-separable manner. we propose Cover-separable Fixed Neural Network Steganography (Cs-FNNS), where a Steganographic Perturbation Search (SPS) algorithm is designed to directly encode the secret data into minor perturbation, which is then transmitted via AI-generated cover images. Such a strategy focuses only on the perturbation encoded from the secret for decoding, eliminating the negative impact of the cover images and favoring the generation of high-quality stego-images.

## 3 THE PROPOSED METHOD

In this section, the proposed Cs-FNNS is elaborated in detail. Following the workflow shown in Fig. 1, we first introduce the Steganographic Perturbation Search algorithm. Then, we present the construction and the sharing of the decoding network. Next, we outline the generation process of the cover images. Finally, we present our solution for hiding multiple images for different receivers.

### 3.1 Steganographic Perturbation Search (SPS)

Let $C \in [0, 1]^{H_c \times W_c \times 3}$ denote an AI-generated RGB cover image with height $H_c$ and width $W_c$. Further, let $S \in [0, 1]^{H_s \times W_s \times 3}$ denote an RGB secret image with height $H_s$ and width $W_s$. Given a decoder $D[\theta] : [0, 1]^{H_c \times W_c \times 3} \rightarrow [0, 1]^{H_s \times W_s \times 3}$ with an architecture $D[\cdot]$ and parameters $\theta$, SPS aims to find a perturbation $\delta \in [0, 1]^{H_c \times W_c \times 3}$

that satisfies the following three properties: 1) minimizing the distance between the cover image $C$ and stego-image $C + \delta$ to ensure visual quality; 2) triggering the decoding network $D[\theta](\cdot)$ to output $S$ to guarantee feasibility; 3) making the stego-image $C + \delta$ deceive steganalysis tools to make sure security. Mathematically, it can be formulated as follows:

$$\min_{\delta} \ dist(C, C + \delta)$$
$$\text{s.t.} \ \begin{cases} D[\theta](\delta) = S \\ J(C + \delta) = 0 \\ 0 \le C + \delta \le 1 \end{cases}, \quad (1)$$

where $dist(\cdot)$ is some distance metrics. *e.g.,* $L_1$, $L_2$ or $L_\infty$ norm, $J(\cdot) = \sum_i j_i(\cdot)$ is a set of steganalysis tools, including statistical tools [3] and recent deep learning-based ones[4, 40, 42], each of which accepts an arbitrary image and output the probability that the image contains secret data. The last box constraint enforces the produced stego-image lies within the normalized pixel space.

The above formulation is difficult to directly solve with existing algorithms, as the constraint $D[\theta](\delta) = S$ and $J(C + \delta) = 0$ are highly non-linear. Therefore, we express them in a different form which is better suited for optimization. We define an objective function $f$ such that $D[\theta](\delta) = S$ and $J(C + \delta) = 0$ if and only if $f(\delta) \le 0$. Here, $f$ is defined as

$$f(\delta) = \|D[\theta](\delta) - S\|_2 + \gamma \sum_i j_i(C + \delta), \quad (2)$$

where $\| \cdot \|_2$ is the $L_2$ norm and $\gamma$ is a hyperparameter that balances the error of the recovered secret image and the undetectability of the stego-image. Here, $j_i(\cdot)$ specifically refers to well-trained deep steganalysis networks. On the one hand, deep steganalysis networks are more powerful than traditional statistical tools and can effectively detect stego-images from the most existing steganographic methods. On the other hand, they are differentiable, which allows us to use their gradient signals to guide the direction of searching $\delta$. Here, we assemble multiple steganalysis networks to enhance the undetectability of $\delta$.

To ensure that the generated stego-image has high visual quality, we jointly use the $L_2$ and $L_\infty$ norms to constrain its distance from the cover image. Now, we reformulate our SPS by

$$\min_\delta \|\delta\|_2 + \beta\, f(\delta)$$

$$\text{s.t.} \quad \begin{cases} \|\delta\|_\infty \le \epsilon \\ 0 \le C + \delta \le 1 \end{cases}, \tag{3}$$

where $\beta$ is a hyper-parameter for balancing the visual quality, the recovery accuracy, and the undetectability. $\|\cdot\|_\infty$ is the $L_\infty$ norm with $\epsilon < 1$ limiting the maximum value of the pixels in $\delta$.

The condition in Eq. 3 imposes two box constraints on $\delta$. Previous FNNS works [23, 30] address them using a particular optimization algorithm, L-BFGS [9], which natively supports box constraints. However, in this paper, we explore a different method of approaching this problem. The two box constraints provide two intervals: $[-\epsilon, \epsilon]$ and $[-C, 1-C]$, which define the admissible values of the pixels in $\delta$. We take the intersection of the two intervals to obtain a more accurate range, ensuring that the searched $\delta$ simultaneously satisfies the two constraints, as follows:

$$\max\{-C, -\epsilon\} \le \delta \le \min\{1-C, \epsilon\}. \tag{4}$$

For simplicity, we denote the lower and upper bounds of $\delta$ as $l = \max\{-C, -\epsilon\}$ and $u = \min\{1-C, \epsilon\}$, respectively. We introduce a new variable $z$ and parameterize $\delta$ below:

$$\delta = l + (u - l)\frac{\tanh(z) + 1}{2}, \tag{5}$$

Since $-1 \le \tanh(z) \le 1$, it follows that $l \le \delta \le u$, so the solution will automatically be valid. Instead of optimizing over $\delta$ defined above, SPS optimizes over $z$ to achieve its goal. That is, given $C$, $S$, $D[\theta](\cdot)$ and $J(\cdot)$, find $z$ that solves

$$\min_z \|\delta\|_2 + \beta\left(\|D[\theta](\delta) - S\|_2 + \gamma\sum_i j_i(C + \delta)\right). \tag{6}$$

The above formulation allows us to use other optimization algorithms that don't support box constraints. We adopt the Adam [21] optimizer to solve it, which is capable of finding effective perturbations quickly.

## 3.2 Decoding Network Construction and Sharing.

Architecture and weights are two essential elements in constructing the decoding network. Previous works [23, 30] have explored the architecture of the decoding networks that are sensitive to perturbation. Based on their findings, we empirically set $D[\cdot]$ a shallow neural network stacked with several convolutional (Conv) layers, Instance normalization (IN) layers, Leaky Rectified Linear Units (LeakyReLU), and a Sigmoid activation function, as shown in Fig. 2. The weights of the Conv layers are four-dimensional tensors, where the first, second, and last two dimensions represent the input channel, output channel, and kernel sizes, respectively. The strides of the Conv layers are variable. By changing them, the size relationship between $\delta/C$ and $S$ can be adjusted, thereby adjusting the embedding capacity.

We set the weights $\theta$ of the decoding network to random values initialized by

$$\theta = \mathcal{I}(D[\cdot], k_d), \tag{7}$$

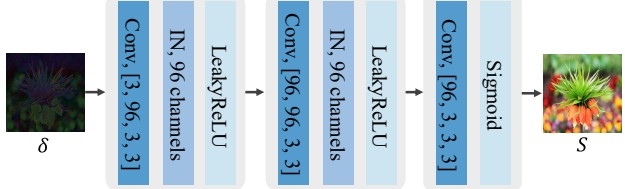

**Figure 2: Architecture of the decoding network.**

where $\mathcal{I}(\cdot)$ is an algorithm for seed based weight initialization. $k_d$ is a key (i.e., seed, not shown in Fig. 1), controlling the initialization process. We empirically explore several initialization algorithms for $\theta$, and evaluate them with respect to the quality of the generated stego-images and recovered secret images, which is detailed in Sec 4.7. Throughout, we use the Xavier algorithm [11], which demonstrates outstanding performance, to initialize $\theta$.

With $D[\cdot]$ and $\mathcal{I}(\cdot)$ being prepared, Alice and Bob are able to construct the same decoding networks using the same $k_d$. Therefore, they only need to share $k_d$ to share the decoding network.

## 3.3 Cover Image Generation

In order to generate the same cover image $C$ for Alice hiding $S$ and Bob recovering $S$, we introduce another key $k_c$ to precisely control the generation of the cover image. Specifically, we set $k_c$ as the seed and sample a noise map from a prior distribution (i.e., Gaussian distribution), which is then fed into a pre-trained deep generative model (say $G$) to produce the cover image by

$$C = G(k_c). \tag{8}$$

## 3.4 Hiding Multiple Images for Different Receivers

Hiding multiple secret images for different receivers in a single cover image is challenging, and only a few work [45] has been able to accomplish it. Here, the difficulties lie not only in the requirement for more embedding capacity, but also in that each receiver must only extract her/his image, and cannot extract (or even perceive the existence of) other images.

The proposed Cs-FNNS provides an elegant extension to hiding multiple images for different receivers, which follows a simple three-step approach. First, construct $T > 1$ different random decoding networks $\{D[\theta_1], \cdots, D[\theta_T]\}$ using $T$ different keys $\{k_{d_1}, \cdots, k_{d_T}\}$, shared to $T$ different receivers, where

$$\theta_t = \mathcal{I}(D[\cdot], k_{d_t}), \ t = \{1, \cdots, T\}, \tag{9}$$

Second, search for a perturbation that could trigger different random decoding networks to output different secret images $\{S_1, \cdots, S_T\}$ by

$$\min_\delta \|\delta\|_2 + \beta\left(\sum_{t=1}^{T} \|D[\theta_t](\delta) - S_t\|_2 + \gamma\sum_i j_i(C + \delta)\right), \tag{10}$$

Third, add the perturbation on an AI-generated cover image $C = G(k_c)$ to generate a stego-image, which is sent to $T$ different receivers through public channels. As such, the $t$-th receiver who possesses $\{k_{d_t}, k_c\}$ could only extract the secret image $S_t$.

Table 1: Visual quality of the stego-images generated using different FNNS schemes, with the best values in bold. "↑": the larger the better, "↓": the smaller the better.

| Methods | Campus-I | | | Campus-II | | | Campus-III | | |
|---|---|---|---|---|---|---|---|---|---|
| | PSNR(dB)↑ | SSIM↑ | LPIPS↓ | PSNR(dB)↑ | SSIM↑ | LPIPS↓ | PSNR(dB)↑ | SSIM↑ | LPIPS↓ |
| Kishore *et al.* [23] | 22.24 | 0.5254 | 0.2087 | 22.28 | 0.5290 | 0.2028 | 21.69 | 0.5018 | 0.2161 |
| Luo *et al.* [30] | 23.42 | 0.5762 | 0.1651 | 23.52 | 0.5825 | 0.1601 | 22.98 | 0.5572 | 0.1701 |
| Ours | **41.89** | **0.9799** | **0.0035** | **42.00** | **0.9804** | **0.0034** | **41.78** | **0.9787** | **0.0034** |

Table 2: Visual quality of the recovered images generated using different FNNS schemes.

| Methods | ImageNet | | | COCO | | | CelebA | | |
|---|---|---|---|---|---|---|---|---|---|
| | PSNR(dB)↑ | SSIM↑ | LPIPS↓ | PSNR(dB)↑ | SSIM↑ | LPIPS↓ | PSNR(dB)↑ | SSIM↑ | LPIPS↓ |
| Kishore *et al.* [23] | 22.79 | 0.7827 | 0.1763 | 23.01 | 0.7970 | 0.1702 | 22.63 | 0.8037 | 0.2033 |
| Luo *et al.* [30] | 21.20 | 0.7413 | 0.2288 | 21.44 | 0.7549 | 0.2223 | 20.98 | 0.7648 | 0.2552 |
| Ours | **33.01** | **0.9156** | **0.0390** | **33.47** | **0.9260** | **0.0376** | **37.04** | **0.9465** | **0.0304** |

## 4 EXPERIMENTS

### 4.1 Experimental Settings

**Datasets.** We take the images from three benchmark datasets as the secret images, including 1000 images randomly selected from the ImageNet validation dataset [34], 1000 images randomly selected from the COCO dataset [27], and 1000 images randomly selected from the CelebA dataset [28]. Before encoding the secret images into perturbations, we resize them to 256 × 256 pixels. We employ a pre-trained Stable Diffusion model [33] as $G(\cdot)$ and use it to construct a cover dataset consisting of 3000 images with the size of 512 × 512. Each of the images in this dataset is generated according to a key with a text prompt of "Campus". The generated cover images are divided equally into three parts, named Campus-I, Campus-II, and Campus-III, for hiding the ImageNet, COCO, and CelebA datasets, respectively. Here, we set the embedding capacity 6 bpp by making the size of the hidden image $S$ one-fourth of the size of $C$ or $\delta$, i.e., $H_c = 2H_s$ and $W_c = 2W_s$. To fit the setting, the strides of the three Conv layers in the decodeing network are set to 1, 1, 2, respectively.

**Optimization** The Adam optimizer [21] with default setting is adopted as the solver to optimize $\delta$. The number of total iterations is 1,500. The initial learning rate is $1 \times 10^{-1.25}$, which is reduced by half every 500 iterations. The perturbation bound $\epsilon$ is set to 0.2. The hype-parameters $\beta$ is set to 0.5, and $\gamma$ is set to 0 for the first 1400 iterations and $2 \times 10^{-5}$ for the last 100 iterations. The well trained SRNet [4] and SiaStegNet [42] steganographic networks are used as $j(\cdot)$ to provide gradient signals for our SPS optimization.

**Evaluation.** There are three metrics utilized to measure the visual quality of the stego-images and the recovered secret images (termed as recovered images for short), including Peak Signal-to-Noise Ratio (PSNR), Structural Similarity Index (SSIM) [37], and Learned Perceptual Image Patch Similarity (LPIPS) [48]. The larger value of PSNR, SSIM and smaller value of LPIPS indicate higher image quality. To highlight the effectiveness of our Cs-FNNS, we compare it with the SOTA FNNS methods [23, 30]. We empirically observe the presence of regular errors (i.e., Gaussian noise) in the recovered images by the existing FNNS schemes. We recommend

the receiver use a lightweight denoising algorithm [47] to post-process their recovery results for better performance (please see supplementary for more details). All the network-generated images are quantified to 8 × 3 bpp before the evaluation.

All our experiments are conducted on Ubuntu 18.04 with an Intel Xeon Silver 4210 CPU and five NVIDIA RTX 2080 Ti GPUs.

### 4.2 Visual Quality

The visual quality of the stego-images generated using different methods is shown in Table 1, where we find that our method significantly outperforms the existing FNNS methods in terms of all the three metrics. Specifically, our Cs-FNNS achieves 18.47 dB, 18.48 dB, and 18.80 dB improvement in PSNR than the second best results on ImageNet, COCO, and CelebA datasets, respectively. In addition to PSNR, similar improvements are also evident in SSIM and LPIPS. Table 2 further presents the performance of different methods in secret image recovery, where can observe that the visual quality of the recovered images by existing FNNS methods is poor (<24dB in PSNR). In contrast, our method demonstrates satisfactory recovery results, surpassing 33dB on the ImageNet and COCO datasets and exceeding 37dB on the CelebA dataset. We obtain significantly better results than the SOTA FNNS methods, thanks to the cover-separability of our Cs-FNNS framework and the proposed SPS algorithm that greatly improve the hiding performance.

Fig. 3 compares the stego and recovered images of our Cs-FNNS and the other two methods (more visual examples are provided in supplementary). As can be seen, in our method, the stego-image is almost identical to its cover version, i.e., the × 10 magnified residual between them is nearly invisible. Moreover, our method offers high recovery accuracy. Its recovery images keep high color fidelity without noticeable artifacts. In contrast, the stego-image in comparison methods shows visible noise, especially in smooth regions. Additionally, their recovery images often suffer from undesirable color deviation problem. In general, these experiments indicate that our method achieves outstanding results both quantitatively and qualitatively.

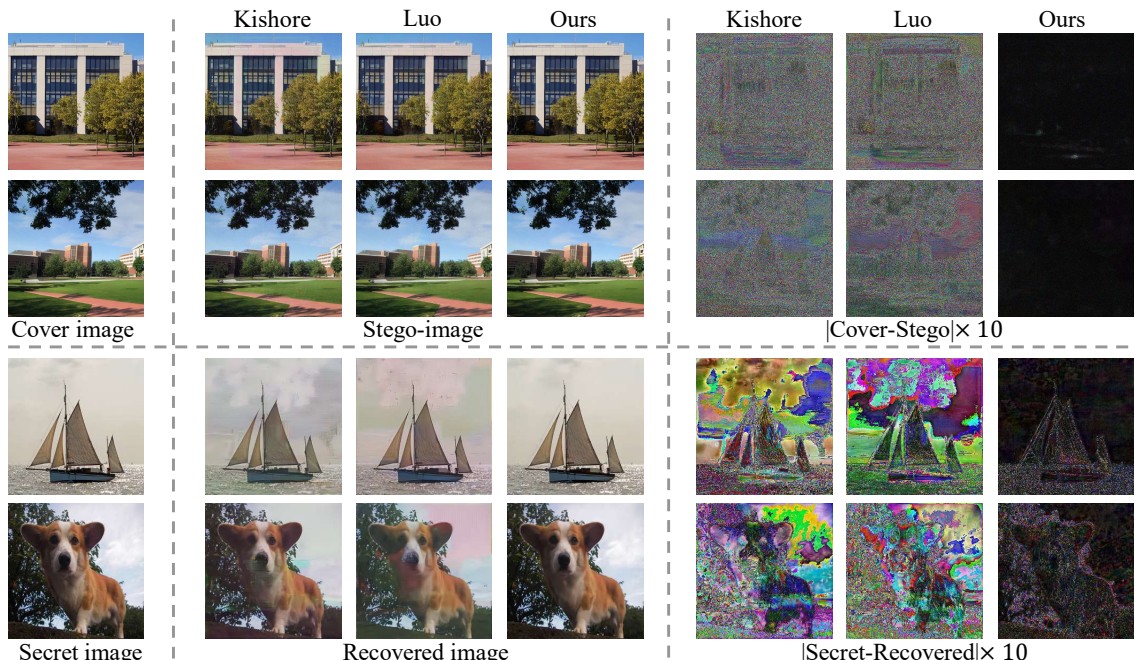

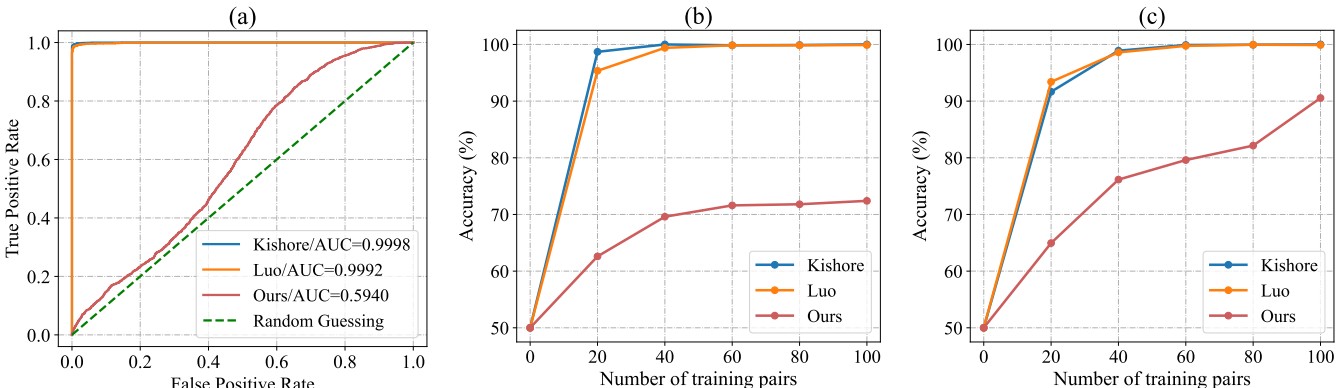

**Figure 3: Visualization of the stego and recovered images generated using different FNNS methods.**

**Figure 4: The undetectability of the stego-images generated using different FNNS methods against (a) StegExpose, (b) YeNet and (c) SiaStegNet.**

## 4.3 Steganalysis

In this section, we evaluate the undetectability of the stego-images generated using different FNNS methods. We adopt three popular image steganalysis tools that are publicly available to carry out the evaluation, including StegExpose [3], YeNet [40] and SiaStegNet [42]. The former is a traditional steganalysis tool which assembles a set of statistical methods, while the latter two are deep learning based steganalysis methods. Using the Campus cover dataset, we generate 3000 stego-images by either our or comparison methods. In other words, there are 3000 cover/stego-image pairs for each FNNS method for evaluation.

**Statistical steganalysis.** By following the protocol in [30] to use the StegExpose, we feed all the cover/stego-image pairs into

the StegExpose for detection. We obtain receiver operating characteristic (ROC) curves for FNNS methods by varying the detection thresholds in StegExpose, which are shown in Fig. 4(a). We would like to mention that the optimal ROC carve for steganography is the diagonal green dotted line and the optimal value of area under ROC carve (AUC) is 0.5, indicating a randomly guessed detection. As can be seen, the AUC of our Cs-FNNS is 0.5940, which is significantly lower than those of the comparison methods, and is close to the optimal value. This indicates the high undetectability of our stego-images against the StegExpose.

**Deep learning based steganalysis.** In order to conduct the evaluation using the YeNet [40] and SiaStegNet [42], we randomly split our cover/stego-image pairs into training (2k pairs) and testing

**Table 3: Visual quality of the stego and recovered images generated by different FNNS methods on the COCO and Campus-II datasets under JPEG compression (quality=90).**

| Methods | Stego-images | | Recovered images | |
|---|---|---|---|---|
| | PSNR ↑ | SSIM↑ | PSNR (dB) ↑ | SSIM↑ |
| Kishore *et al.* [23] | 28.53 | 0.7830 | **15.11** | **0.4450** |
| Luo *et al.* [30] | 29.17 | 0.8046 | 14.54 | 0.4106 |
| Ours | **41.95** | **0.9827** | 12.20 | 0.2786 |
| ★ Kishore *et al.* [23] | 17.89 | 0.4023 | 17.13 | 0.5789 |
| ★ Luo *et al.* [30] | 10.68 | 0.1047 | 20.44 | 0.7018 |
| ★ Ours | **24.80** | **0.6300** | **29.18** | **0.8742** |

(1k pairs) parts. By following the protocol given in [13, 20], we train the two steganalysis networks from scratch with cover/stego-image pairs of our training part. Specifically, we gradually increase the number of training samples to investigate how many image pairs are needed to make the steganalysis networks capable of detecting the stego-images in our testing part. We also test the comparison methods under the same setting as ours. Fig. 4(b) and Fig. 4(c) depict the detection accuracy of the YeNet and SiaStegNet under different number of training image pairs. We can see that our Cs-FNNS achieves much lower detection accuracy compared to other methods, which shows the higher undetectability of our method.

## 4.4 JPEG Compression

JPEG compression is an effective way to remove the hidden data from the stego-images. Existing steganographic techniques, especially high-capacity ones, struggle to resist it. In this section, we evaluate the ability of the existing FNNS methods and our Cs-FNNS against JPEG compression. To improve the robustness of the methods, we add a JPEG layer (with a quality factor of 90) before their decoding networks during the optimization phase. Due to the non-differentiability of the JPEG layer, we approximated its back-propagated gradient with identity transformation [46]. For clarity, we use ★ to mark the adapted methods. We also actively reduced the payload to achieve higher performance. In particular, we adjust the embedding capacity to 1.5 bpp, and hide the downsampled secret image with a resolution of $128 \times 128$ into cover images sized at $512 \times 512$ pixels. Correspondingly, the strides of the last two Conv layers in the decoding networks are set to 2.

Table. 3 presents the quality of the stego-images and recovered images generated using different FNNS methods under JPEG compression. As can be seen, without the adaptive JPEG layer assisting optimization, FNNS methods can obtain high-quality stego-images. However, once the stego-images are compressed, their secret recovery accuracy significantly decrease. Among them, our Cs-FNNS is the most affected and recovers the lowest-quality hidden images (e.g., 12.20dB). This is because our method creates the stego-images with smaller distortion, which is more susceptible to being erased by JPEG compression. Compared to the original FNNS methods, the adapted versions successfully find perturbations that won't be lost under the JPEG compression, thereby maintaining a reliable quality for secret image recovery. Among of the adapted methods, our Cs-FNNS achieves the best secret recovery accuracy (PSNR ≥ 29dB).

**Table 4: Comparison on computational efficiency.**

| Methods | Embedding | Extraction |
|---|---|---|
| Kishore [23] | 12.59 | 0.06 |
| Luo [30] | 11.54 | 0.06 |
| Ours | 30.02 | 12.08 |

Additionally, we also provide the highest quality stego-images. In general, these results highlight the superior anti-JPEG ability of our method over the previous FNNS methods.

## 4.5 Computational Efficiency

Cs-FNNS involves computation in two aspects: Cover Image Generation (CIG) and Steganographic Perturbation Search (SPS). For CIG, we utilize the built-in code of PyTorch framework to run pre-trained Stable-Diffusion-v1-5, which can only generate images of size $512 \times 512$. The average running time (in seconds) for generating a cover image is 11.96s. For SPS, we resize the generated cover images to different resolutions and run SPS on them. The average computational time for searching $\delta$ for cover images $C$ with $256 \times 256$, $512 \times 512$, and $1024 \times 1024$ resolution are 6.55s, 18.06s and 73.18s, respectively. We can observe that the optimization time scales approximately linearly with the number of pixels in $C$. For every fourfold increase in the number of pixels, the time increases by a little less than a factor of 4.

On cover images with a resolution of $512 \times 512$, we compare our method with the existing SOTA FNNS methods [23, 30] in terms of computational efficiency. It should be noted that previous FNNS methods do not involve the CIG. When embedding, they individually optimize a cover image towards a fixed decoding network to generate a stego-image; when extraction, they feed the stego-image into the same decoding network to recover the secret data. The average computation times for different stages of the FNNS methods are given in Table 4, where only our embedding and extraction results include the time for CIG. We can see that our method shows lower computational efficiency compared to the other two methods, especially in the extraction stage. However, neither the embedding nor extraction times of our Cs-FNNS exceed 40 seconds, which is adequate for use in the majority of real-world steganographic applications.

## 4.6 Hiding Multiple Images for Different Receivers.

In this section, we evaluate the performance of our method in hiding multiple images for different receivers. We compare our Cs-FNNS with the UDH [45], which, to the best of our knowledge, is the SOTA method in this field. UDH trains multiple decoding networks to extract different secret images from a stego-image. Due to the limited computational resources, we train it to hide up to four secret images. *i.e.*, $T \in \{2, 3, 4\}$.

Table. 5 summarizes the hiding results of UDH and our Cs-FNNS on the ImageNet and Campus-I datasets. We can see that the stego-images generated by the two methods have their own strength and weakness. Compared to UDH, Cs-FNNS produces stego-images with lower pixel errors but poorer perceptual similarity. However,

**Table 5: Visual comparisons between Cs-FNNS and UDH in terms of hiding multiple images for different receivers**

| Number of secret images | UDH [45] | | | | | | Cs-FNNS | | | | | |
| | Stego-images | | | Recovered images | | | Stego-images | | | Recovered images | | |
| | PSNR(dB)↑ | SSIM↑ | LPIPS↓ | PSNR(dB)↑ | SSIM↑ | LPIPS↓ | PSNR(dB)↑ | SSIM↑ | LPIPS↓ | PSNR(dB)↑ | SSIM↑ | LPIPS↓ |
|---|---|---|---|---|---|---|---|---|---|---|---|---|
| Two | 34.86 | 0.9544 | **0.0009** | 27.40 | 0.7959 | 0.1390 | **40.16** | **0.9715** | 0.0049 | **31.38** | **0.8989** | **0.0234** |
| Three | 34.76 | **0.9633** | **0.0031** | 26.88 | 0.8011 | 0.1422 | **37.54** | 0.9492 | 0.0104 | **28.40** | **0.8739** | **0.0207** |
| Four | 32.70 | **0.9436** | **0.0127** | 21.63 | 0.7020 | 0.2358 | **34.47** | 0.9104 | 0.0207 | **25.02** | **0.8370** | **0.0239** |

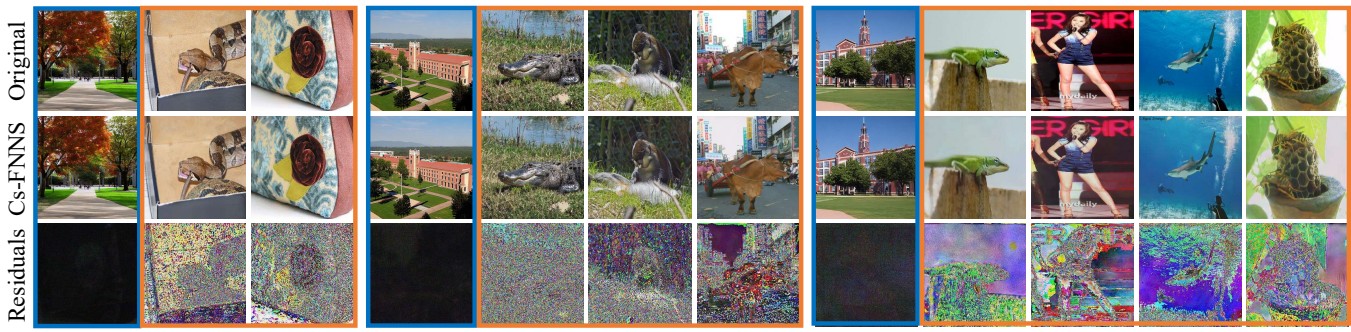

(a) 2 hidden images      (b) 3 hidden images      (c) 4 hidden images

**Figure 5: Visualization of our Cs-FNNS when hiding multiple secret images for different receivers. Sub-figure (a), (b), and (c) respectively depict the results of hiding 2~4 secret images, with a blue border on the cover/stego images and an orange border on the secret/recovered images. In each sub-figure, the top row gives the original images and the middle row displays our generated results, while the third row shows the × 10 magnified residuals between them.**

in secret recovery, regardless of the number of hidden images, Cs-FNNS outperforms UDH in all three metrics. Specifically, when hiding four secret images, our recovery images have 3.39dB and 0.21 advantages over those of UDH in PSNR and LPIPS, respectively.

Fig. 5 visualizes our results for hiding 2~4 secret images, with a blue border around cover/stego images and an orange border around the secret/recovered images. We can observe that our Cs-FNNS produces satisfactory stego-images, i.e., the × 10 magnified residuals between the cover and stego-images are nearly imperceptible. Despite the presence of noise in the recovered images, their main content remains clearly visible. This indicates the effectiveness of Cs-FNNS in hiding multiple images for different receivers.

## 4.7 Weight Initialization.

For the convenience of sharing the decoding network between the sender and receiver, we set the weight $\theta$ of the decoding network to randomly initialized values. On the ImageNet and Campus-I datasets, we empirically explore several algorithms [11, 15, 35] for initializing $\theta$, and evaluate them w.r.t. the visual quality of the generated stego-images and recovered secret images.

Table 6 summaries the performance of the decoding networks with random weights initialized by different algorithms, where $\mathcal{U}(0, 1)$ and $\mathcal{N}(0, 1)$ denote that $\theta$ are sampled from the Uniform distribution and Standard Gaussian distribution, respectively. We can observe that the Xariver, Orthogonal, and Kaiming initialization algorithms [11, 15, 35] achieve nearly identical outstanding performance, significantly surpassing the first two algorithms. Therefore, we adopt the Xavier algorithm to initialize $\theta$ in our experiments.

**Table 6: Performance of the decoding networks with random weights initialized by different algorithms.**

| Initialization Algorithms | Stego-images | | Recovered images | |
| | PSNR(dB)↑ | SSIM↑ | PSNR(dB)↑ | SSIM ↑ |
|---|---|---|---|---|
| $\mathcal{U}(0, 1)$ | 29.87 | 0.9458 | 8.61 | 0.1564 |
| $\mathcal{N}(0, 1)$ | 27.84 | 0.7419 | 14.53 | 0.3439 |
| Xavier [11] | 41.92 | 0.9807 | 32.77 | 0.9183 |
| Orthogonal [35] | 42.09 | 0.9808 | 32.22 | 0.9130 |
| Kaiming [15] | 41.84 | 0.9805 | 32.80 | 0.9186 |

## 5 CONCLUSION

In this paper, a Cover-separable FNNS scheme is proposed with the support of the deep generative models for cover image generation. Unlike previous FNNS methods, Cs-FNNS separates the decoding of the secrets from the cover image, eliminating disturbance from the cover image in data extraction, hence achieving a better performance. To align with this novel decoding way, we propose a steganographic perturbation search (SPS) algorithm to directly encode the secret images into imperceptible perturbations. By using SPS, we successfully find the perturbations that trigger up to four different random decoding networks to output different secret images. The perturbations are minimal and negligible, ensuring both the visual quality and undetectability of the stego-images. Experimental results demonstrate the superiority of the proposed method to the existing FNNS schemes.

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
