# OpenReview forum: "Cover-separable Fixed Neural Network Steganography via Deep Generative Models"
_acmmm.org/ACMMM/2024/Conference — MM2024 Poster_

### Official Review · Reviewer_vWdp · 2024-05-13

**Rating:** 3
**Confidence:** 4

**Summary:**

This paper proposes a cover-separable fixed neural network steganography (Cs-Fixed FNNs) method to encode the secret data into an imperceptible perturbation that is searched by the devised Steganographic Perturbation Search (SPS) algorithm, which is then added into the AI-Generated cover image generated by the specific key for transmission. During the decode stage, the receiver uses the same key to spawn the same cover image and then generates the perturbation that is used to recover the secret image. The author conducts extensive experiments to verify their method's effectiveness.

**Strengths:**

1. The author devises a perturbation search algorithm to convey the secret image to imperceptible perturbation, which is then severed as the secret container, implementing an invisible data hiding. Meanwhile, the secret images can be decoded from perturbation via a fixed random decoding network.
2. The author conducts extensive experiments to prove the effectiveness of their method to comparison methods in terms of several metrics.

**Limitations:**

1. The author overclaims that their method does not require training, as an important component of their method, the AI-generated cover image via a generative model also requires extensive training to engender high fidelity image.
2. Given the randomness of the AI-generated network, how can you ensure the receiver can use the key to generate the cover image accurately?
3. The introduction of Section 3.2 is too simple; if the network does not require training, only using random initialization of a simple network can generate a high-fidelity image. It seems hard to generate a high-fidelity image as the paper reported.
4. How much secret image does the Cs-FNNS support in the setting of hiding multiple images?
5. More discussion on gamma and beta is required.
6. Is the optimized perturbation universal?  or does each cover or secret have to optimize a new perturbation?
7. From Table 3, the proposed method is susceptible to the distortion. More distortion should be considered, such as Gaussian blur.
8. The notion of D[θ](·) is queer, is the θ is the model parameter, the common notion is D_θ(·) or D(·; θ)
9. Although the proposed method is based on fixed neural networks, a comparison with some state-of-the-art works based on learned neural network steganography is required, which makes your work more convincing.
10. Is the steganalysis network used to train the perturbation the same as the evaluation one? If so, it may be unfair to conduct an evaluation as the adversarial training would invalidate the detection ability of the steganalysis network. The author should use different steganalysis networks to perform evaluation.

**Suitability:**

2

---

### Official Review · Reviewer_hZLq · 2024-05-18

**Rating:** 4
**Confidence:** 3

**Summary:**

This paper proposed Cs-FNNS, which used the fixed neural network to hide and extract secret images.

**Strengths:**

1. This paper is easy to follow. The structure is clear and reasonable. The figures and tables are informative and easy to understand.
2. The principle of using a fixed neural network for steganography is easy and effective.
3. The proposed Cs-FNNS is more imperceptible than previous work, and has better extraction accuracy.
4. The Cs-FNNS has some robustness to JPEG compression (QF=90).

**Limitations:**

1. The steganography analysis models are included in the training of Cs-FNNS. So what about the imperceptibility of Cs-FNNS when it is detected by an unseen steganography analysis model?
2. The selected baselines are somewhat old-fashioned, especially the Kishore (2021). The reviewer expects the authors to select the state-of-the-art methods to compare (such as but not limited to [1]).
3. The robustness to JPEG compression is important for image steganography applications, and the QF=90 is too high. The authors are expected to verify the robustness of Cs-FNNS in severe distortion compression (lower QF).


[1] CRoSS: Diffusion Model Makes Controllable, Robust and Secure Image Steganography, NIPS 2023

**Suitability:**

3

---

### Official Review · Reviewer_mmBp · 2024-05-24

**Rating:** 3
**Confidence:** 4

**Summary:**

This paper proposes a cover-separable fixed neural network steganography. A steganographic perturbation search algorithm is presented to encode the secret data into an imperceptible perturbation, which is combined with an Al-generated cover image for transmission. The experiments demonstrate its superior performance over previous fixed neural network steganography. However, the major concern is that some key experiments are missing.

**Strengths:**

1) The paper in general is well written and well organized.
2) The technical details are solid.
3) The experimental results are compelling.

**Limitations:**

1. The fixed neural network steganography proposed in this paper is characterized by the separable cover. However, there is a lack of explanation regarding what cover-separable means and the motivation for adopting a cover-separable approach. The authors are advised to provide a detailed explanation and justification for this choice.

2. The steganographic perturbation is generated by a steganographic perturbation search algorithm and is independent of the cover image. Does this imply that the steganographic perturbation could be universal, similar to the UDH method? The authors should conduct additional experiments to demonstrate this potential universality.

3. The robustness experiments in the paper only consider JPEG compression. Since steganographic perturbation is a form of noise, it is necessary to evaluate the robustness of the steganographic method against noise attacks. The authors should include experiments that assess the steganography's resilience to various noise attacks.

**Suitability:**

2

---

### Official Review · Reviewer_DZtv · 2024-05-25

**Rating:** 4
**Confidence:** 4

**Summary:**

This paper proposes a fixed neural network steganography scheme with separable cover images by means of a deep generative model capable of generating cover images. With this scheme, the Cs-FNNS proposed in this paper is able to completely separate the cover image from the secret image, which is the main reason why this method shows excellent performance. In order to imperceptibly embed the secret image into the cover image, this paper proposes a steganographic perturbation search algorithm. Furthermore, the program can be easily extended to hide multiple images for different recipients.

**Strengths:**

1. Cs-FNNS can completely separate the cover image from the secret image with the help of a deep generative model, which can eliminate the influence of the cover image under ideal circumstances.
2. This paper proposes a steganographic perturbation search algorithm that directly encodes the secret image into an imperceptible perturbation. The decoder, steganalysis tool, and visual quality are taken into account in the steganographic perturbation search.
3. The program can be easily extended to hide multiple images for different recipients.

**Limitations:**

1. When decrypting the solution proposed in the paper, the receiver needs to know the keys used by the sender to construct the overlay image for a specific overlay image. If different secret images need to be transmitted multiple times, a lot of communication is required between the two parties to negotiate the keys and corresponding overlay images. The paper should describe in detail the process of how the receiver communicates and negotiates the keys of the steganographic image with the sender after obtaining a steganographic image. Is it possible to embed the keys into the overlay image and the receiver extracts the keys first and then the secret image?
2. The disadvantage of learning neural network steganography introduced in the paper is that it requires training and transmission, and the deep generative model in the solution also requires training and transmission. If a public pre-trained model is used, an attacker can easily obtain the Perturbation in the scheme by intercepting the keys.
3. For a single process of secret image embedding and extraction, it is challenging to achieve practical application because the sender and receiver need to reach a consensus on three values: the key (kc) for generating the random cover image, the random seed (kd) for generating the fixed random decoding network parameters, and the parameters (G) of the deep generative model.

**Suitability:**

2

---

### Meta-Review · Area_Chair_YDJ9 · 2024-07-04

**Recommendation:** Accept (Poster)
**Confidence:** 4

**Metareview:**

The manuscript received four reviews, based on which a rebuttal was submitted.

In the final rating, three reviewers are on the positive side, while the last one gave borderline reject.

Essentially, the reviewers are happy with the technicals aspects, the experiments, and the overall quality of the manuscript.

As such, the AC follows the consensus and recommends the acceptance.

Please, however, do account for the comments and revise in the final version.